# Design for Interpretability

**Anagha Kulkarni**[1][*] · **Sarath Sreedharan**[1][*] · **Sarah Keren**[2] · **Tathagata Chakraborti**[3]
**David E. Smith** · **Subbarao Kambhampati**[1]

[1]Arizona State University
[2]Harvard University
[3]IBM Research AI

*anaghak@asu.edu, ssreedh3@asu.edu, skeren@seas.harvard.edu, tchakra2@ibm.com, david.smith@psresearch.xyz, rao@asu.edu*

## Abstract

The interpretability of an AI agent's behavior is of utmost importance for effective human-AI interaction. To this end, there has been increasing interest in characterizing and generating interpretable behavior of the agent. An alternative approach to guarantee that the agent generates interpretable behavior would be to design the agent's environment such that uninterpretable behaviors are either prohibitively expensive or unavailable to the agent. To date, there has been work under the umbrella of *goal or plan recognition design* exploring this notion of environment redesign in some specific instances of interpretable of behavior. In this position paper, we scope the landscape of interpretable behavior and environment redesign in all its different flavors. Specifically, we focus on three types of interpretable behaviors – explicability, legibility and predictability – and present a general framework for environment design that can be instantiated to achieve these behaviors. We also discuss how specific instantiations of this framework correspond to prior works on environment design, and identify exciting opportunities for future work.

## 1 Introduction

The design of human-aware AI agents must ensure that its decisions are interpretable to the human in the loop. Uninterpretable behavior can lead to increased cognitive load on the human – from reduced trust, productivity to increased risk of danger around the agent (Fan et al. 2008). Christensen et al. (2009) emphasises in the *Roadmap for U.S. Robotics – "humans must be able to read and recognize agent activities in order to interpret the agent's understanding"*. The agent's behavior may be uninterpretable if the human: (1) has incorrect notion of the agent's beliefs and capabilities (Zhang et al. 2017; Chakraborti et al. 2017b; Kulkarni et al. 2019) (2) is unaware of the agent's goals and rewards (Dragan and Srinivasa 2013; Kulkarni, Srivastava, and Kambhampati 2019) (3) cannot predict the agent's plan or policy (Fisac et al. 2018; Kulkarni, Srivastava, and Kambhampati 2019). Thus, in order to be interpretable, the agent must take into account the human's expectations of its behavior – i.e. *the human mental model* (Chakraborti et al. 2017a). There are many ways in which considerations of the human mental model can affect agent behavior.

---

[*]equal contribution

### 1.1 The Landscape of Interpretable Behavior

There has been significant interest recently in characterizing different notions of interpretable behavior of a human-aware AI agent (Chakraborti et al. 2019). Three important properties of interpretable behaviors emerge, namely – (1) *explicability*: when the agent behavior conforms to the expectations of the human; (2) *legibility*: when the agent behavior reveals its objectives or intentions to the observer; and (3) *predictability*: when the (remaining) agent behavior can be precisely predicted by the human.

In existing works, the generation of these behaviors was explored from the point of view of the agent – i.e. the agent altered its planning process by using its estimation of the mental model of the human in the loop in order to exhibit the desired behavior. We refer the reader to (Chakraborti et al. 2019) for a detailed treatise of these concepts.

### 1.2 Environment Design

A parallel thread of work, under the umbrella of *goal and plan recognition design*, has looked at the notion of changing the environment of an agent to increase interpretability of the behaviors available to an agent. The design of environment can be optimized in order to maximize (or minimize) some objective for the actor (for example, optimal-cost to a goal, desired behavioral property) (Zhang, Chen, and Parkes 2009; Keren et al. 2017). An environment design problem takes the initial environment configuration as input, along with set of modifications allowed in the environment and outputs a sequence of modifications, which can be applied to the initial environment to derive a new environment in which the desired objective is optimized. The problem of environment design is more suited for structured settings where an actor performs repetitive objectives (for example, on factory floors, etc). It is also suited for settings involving multiple actors, where the expectations of the observers are the same but there are multiple actors in the environment, making environment design an effective choice (for example, in restaurants with waiter robots where the human customers have same expectations).

**Goal (and Plan) Recognition Design**   In existing work, the concept of environment design for planning agents has been studied in the concept of goal (or plan) recognition

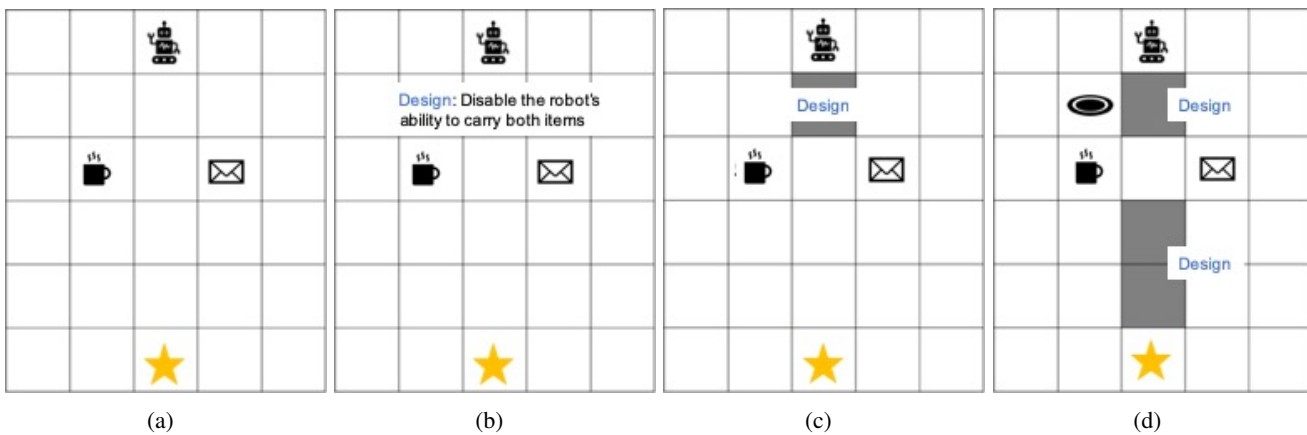

Figure 1: The office assistant domain: (a) The original domain; (b) The domain can be updated for more explicable behavior by disabling the robot coffee holder; (c) To induce legible behavior, we can add dividing walls to constrain the agent and help the observer reduce uncertainty in their mental model; and (d) To induce predictable behavior we can reduce uncertainty about the order of pickup by including a tray that allows the agent to pick up the objects in any order.

design (Keren, Gal, and Karpas 2014; Mirsky et al. 2019) in order to make the goals (or plans) of an actor easier to recognize to the observer. Immediately, this should remind the reader of *legibility* (and *predictability*) introduced above. The goal of this paper is to bridge the gap between these two parallel threads of work and explore the full spectrum of interpretable behavior in the context of environment design. We will discuss how existing work in environment design fits into this narrative and highlight much needed gaps in existing work as exciting avenues for future work.

### 1.3 Why Design for Interpretability

Adopting the reasoning capabilities of an agent to deal with the human mental model, as done in classical human-aware decision-making versus the design of environments are indeed complimentary approaches towards achieving the same purpose: behavior of the agent that is more interpretable to the observer. While one could conceive of the most general framework that accounts for both, it is useful to recognize that these have their own unique set of features. Perhaps, the biggest advantage of environment design is that the process of generation of interpretable behavior is offloaded from the actor onto the design process. In other words, the computational overhead of generating interpretable behavior – in having to deal with the human mental model and reason in the space of models – is now part of the design process only, which can be done offline.

We will see later, in our formulation, how the actor in the "Design for Interpretability" is modeled as a cost-optimal agent that is able to produce interpretable behavior while still planning in the traditional sense. In addition, design also means that the observer does not have to count on the agent to be cooperative and interpretable, and instead can deal with adversarial agents as well. At the end of this paper, we will see that this advantage does come with a caveat.

In general, the notion of interpretable behavior is complemented by communication: e.g. authors in (Chakraborti,

Sreedharan, and Kambhampati 2019) balance considerations of explanations and explicability, while authors in (Chakraborti et al. 2018) balance out intention projection actions for the sake of legibility and predictability. Communication operates in model space by being able to change the observer's beliefs. Design, also operating in model space, can be seen as an alternative to communication that complements the notion of interpretable behavior.

### 1.4 An Illustration of Design for Interpretability

Consider an office setting where an office assistant robot, responsible for delivering items such as coffee or mail to the employees, is about to be deployed (Figure 1a). The robot (*actor*) will be supervised by office security guards (*observer*) who have worked with previous generation office assistant robots and have some expectations regarding their functions. In particular, they expect the robot to carry one item at a time (i.e. either mail or coffee) and each robot generally has a strong preference on the order in which it picks up these items (though the order changes from robot to robot). Unknown to the guards, the new model adds more flexibility to the robot by (1) removing the need for the robots to have fixed preference on the order to pick up items and (2) installs a coffee cup holder that allows the robot to carry both mail and coffee at the same time. Now if we allow the new robot to simply act optimally in the original setting, it would unnecessary confuse the observers.

If the robot was built to generate interpretable behavior, it will change its behavior (and possibly settle for suboptimal decisions in its own model) in order to conform to expectations or it will provide explanations that address these model differences. However, the same effect can be achieved if the designers who are *deploying* the robot also designed the environment to ensure that decisions of the new robot remain interpretable to the occupants of the office.

If the designers wish to prioritize explicability, then the change that they would need to make would be to disable the

coffee holder, this will cause the robot to choose one of the items first, deliver it and then move on to the second one. For explicability, it does not matter which one the robot chooses as the user would simply assume that the order chosen by the robot is the one enforced by the robot's model. As for legibility, the aim is to help the user differentiate between the models as early as possible, one way to do it would be to disable the coffee holder and then build introduce obstacles as shown in Figure 1c. Finally, for predictability, the focus is to allow the user to be able to predict the entire plan as early as possible. One possible design for this scenario is to disable the coffee holder and provide the robot with a tray that allows the robot to carry both items at the same time. The observer can see the tray and realizes the robot can place both items in the tray and the order of picking up no longer matters. In predictability, we may need to add additional obstacles to further restrict the space of possible plans that can be done by the robot (Figure 1d).

## 2 Background

An interpretable decision making problem involves two entities: an actor ($A$) and an observer ($O$). The actor operates in an environment while being observed by the observer.

**Definition 1.** *An **interpretable decision making problem** is a tuple, $\mathcal{P}_{Int} = \langle P_A, \mathbf{P_O}, Int \rangle$, where:*

- *$P_A$ is the decision making problem of the actor $A$*

- *$\mathbf{P_O} = \{P_O^i\}$ is observer's mental model of the actor, represented by a set of possible decision making problems that the observer believes that the actor may be solving.*

- *$Int : \Pi \to \mathbb{R}$ is the interpretability score that is used to evaluate agent plans (where $\Pi$ is the space of plans)*

Interestingly, we do not require that $P_A \in \mathbf{P_O}$ – i.e. the problems in $\mathbf{P_O}$ can be different from $P_A$ in all possible aspects (e.g. state space, action space, initial state and goals). The solution to $\mathcal{P}_{Int}$ is a plan or policy that not only solves $P_A$ but also satisfies some desired properties of interpretable behaviors (measured through the interpretability score). The score could reflect properties like explicability, legibility or predictability of the plan.

**Explicability** The actor's behavior is considered explicable if it aligns with at least one of the observer's expected plans (as per their mental model). The set of plans expected by the observer consists of all the cost-optimal solutions for problems in $\mathbf{P_O}$. The target of explicability is thus to generate behavior that belongs to this set of expected plans.

**Legibility** With legibility, the objective of the actor is to inform the observer about its model – i.e. reduce the size of $\mathbf{P_O}$. An actor's behavior is said to be perfectly legible if it can be derived from only one model in $\mathbf{P_O}$. The longer it takes for a plan prefix to achieve this, the worse is the plan's legibility. This notion of interpretability thus helps the observer narrow down their belief over the possible actor models as quickly as possible.

**Predictability** The objective of the actor with predictability is to generate the most disambiguating behavior – i.e. given the actor's plan prefix, the observer should be able to predict its completion. These predictions would be in terms of cost-optimal completions of a given prefix in the possible problems in the mental model. This means that if there exists the same unique completion in all of the models then *the plan is predictable even though not legible*. The shorter the length of the disambiguating plan prefix, the better the predictability of the plan. An empty prefix would thus correspond to the most predictable plan.

## 3 Design for Interpretability

In this section, we present a general formulation for the design problem for interpretable behaviors. Given an environment design, we assume that the actor is a rational agent and therefore is incentivized to generate cost-optimal plans. Let the set of cost optimal plans of the actor be $\Pi_{P_A}^*$. A cost-optimal plan solution to $P_A$ can exist anywhere on the spectrum of interpretability from high to low. Therefore, we need a measure to quantify the interpretability score for the actor's set of cost-optimal plans. To that end, we introduce the worst-case interpretability score $wci$ as follows:

**Definition 2.** *The **worst-case interpretability score** $wci(\cdot)$, for $\mathcal{P}_{Int}$ is defined as*

$$wci(\mathcal{P}_{Int}) = \min_{\pi \in \Pi_{P_A}^*} Int(\pi) \qquad (1)$$

$Int(\cdot)$ is instantiated for each type of interpretable behavior separately and is discussed in detail at the end of this section. The higher the interpretablity score, the better the interpretability of the behavior (in terms of either of three properties). Therefore, the worst-case interpretability score is the minimum interpretability score of a cost-optimal plan of the actor.

We can now define the design problem for interpretability. When a modification is applied to the environment, both the actor's decision making problem and the observer's mental model are modified, thereby changing the worst-case interpretability score of the actor's cost-optimal plans for the given decision making problem. Let $\mathcal{P}$ denote the set of valid configurations in the real environment. Although $P_A \in \mathcal{P}$, problems in $\mathbf{P_O}$ might not necessarily be in $\mathcal{P}$ if the observer has incorrect or infeasible notions about the actor's model. Therefore, we represent the set of configurations that the observer thinks are possible as $\widetilde{\mathcal{P}}$, and $\mathbf{P_O} \subseteq \widetilde{\mathcal{P}}$.

**Definition 3.** *The **design problem for interpretability**, DP-Int, is a tuple $\langle \mathcal{P}_{Int}^0, \Delta, \Lambda_A, \Lambda_O \rangle$ where,*

- *$\mathcal{P}_{Int}^0 = \langle P_A^0, \mathbf{P_O}^0, Int \rangle$ where $P_A^0 \in \mathcal{P}$ and $\mathbf{P_O}^0 \subseteq \widetilde{\mathcal{P}}$ are the initial models.*

- *$\Delta$ is the set of modifications that can be applied to the environment. $\xi$ is a sequence of modifications.*

- *$\Lambda_A : \Delta \times \mathcal{P} \to \mathcal{P}$ and $\Lambda_O : \Delta \times \widetilde{\mathcal{P}} \to \widetilde{\mathcal{P}}$ are the model transition function that specify the resulting model after applying a modification to the existing models.*

The set of possible modifications includes modifying the set of states, action preconditions, action effects, action

costs, initial state and goal. Each modification $\xi \in \Delta$ is associated with a cost, such that, $C(\xi) = \sum_{\xi_i \in \xi} C(\xi)$. After applying $\xi$ to both $P_A^0$ and $\mathbf{P_O}^0$, the resulting actor decision making problem model and observer mental model are represented as $P_A^{|\xi|}$ and $\mathbf{P_O}^{|\xi|}$ respectively.

Let $\mathcal{P}_{Int}^{|\xi|}$ be the modified interpretable decision making problem after applying the modification $\xi$ to $\mathcal{P}_{Int}$. Our objective here is to solve *DP-Int* such that the worst-case interpretability score of $\mathcal{P}_{Int}$ is maximized. Apart from that, the design cost of $\xi$ has to be minimized, as well as the cost of a plan $\pi_A$ that solves $P_A^{|\xi|}$.

**Definition 4.** *A **solution to** **DP-Int**, is a sequence of modifications $\xi$ with*

$$\min(-wci(\mathcal{P}_{Int}^{|\xi|}), C(\xi), cost(\pi_A)) \qquad (2)$$

This completes the general framework of design for interpretability. In the following, we will look at specific instances of design for the different notions of interpretability.

### 3.1 Design for Explicability

In order to be explicable, the actor's plan has to be consistent with the observer's expectations of it. The observer has an implicit assumption that the actor is a rational agent. Therefore the set of plans expected by the observer includes the cost-optimal plans for all the planning models in the observer's mental model. Let $\Pi_{\mathbf{P_O}}^*$ be the set of expected plans for the observer. Given the set of expected plans, the explicability of the actor's plan depends on how different it is from the expected plans. In order to quantify the explicability of a plan, we introduce the following scoring function:

**Definition 5.** *The **explicability score** $Exp(\cdot)$ of an actor's plan $\pi_A$ that solves $P_A$ is defined as follows:*

$$Exp(\pi_A) = \max_{\pi \in \Pi_{\mathbf{P_O}}^*} e^{-\delta_{\mathbf{P_O}}(\pi_A, \pi)} \qquad (3)$$

Here $\delta_{\mathbf{P_O}}(\cdot)$ computes the distance between two plans with respect to the observer's mental model. For example, the distance function could compute a cost-based difference between the two plans in the observer's mental model. Plugging this scoring function in Equation 2 allows us to instantiate the design problem for explicability.

### 3.2 Design for Legibility

In order to be legible, the actor's plan has to reveal its problem to the observer as early on as possible. Therefore, the legibility of a plan is inversely proportional to the length of its shortest prefix that has unique cost optimal completion for more than one problem in the observer's mental model.

**Definition 6.** *The **legibility score** $Leg(\cdot)$ of an actor's plan, $\pi_A$, that solves $P_A$ is defined as follows:*

$$Leg(\pi_A) = \min_{\tilde{\pi}_A \in \tilde{\Pi}_{P_A}} e^{-|\tilde{\pi}_A|} \qquad (4)$$

such that $\exists (P_O^i, P_O^j) \in \mathbf{P_O}, i \neq j$ with unique cost optimal completion of $\tilde{\pi}_A$ in each model, and $\tilde{\Pi}_{P_A}$ is the set of all prefixes of $\pi_A$. Plugging this scoring function in Equation 2 allows us to instantiate the design problem for legibility.

### Goal Recognition Design
The work on goal recognition design (GRD) (Keren, Gal, and Karpas 2014) is a special case of the design problem for legibility. The GRD problem involves an actor and an observer where the observer's mental model consists of planning models that have the exact same state space, actions and initial state as the actor's planning model. However, each planning model in the observer's mental model has a different goal. The actor's true goal is one of them, and the objective of GRD problem is to redesign the environment, such that, the true goal of the actor is revealed to the observer as early as possible. The interpretability problem defined here is a general one, where the observer's mental model can be different in all possible ways from the actor's actual planning model.

### 3.3 Design for Predictability

In order to be predictable, the plan has to be the most-disambiguating plan among the set of plans the observer is considering – i.e. the observer should be able to predict the rest of the plan after seeing the prefix. Therefore, predictability of a plan is inversely proportional to the length of its shortest prefix which ensures only one optimal completion solving only a single problem in the observer's mental model. We can quantify the predictability score as follows:

**Definition 7.** *The **predictability score** $Pred(\cdot)$ of an actor's plan $\pi_A$ that solves $P_A$ is defined as follows:*

$$Pred(\pi_A) = \min_{\tilde{\pi}_A \in \tilde{\Pi}_{P_A}} e^{-|\tilde{\pi}_A|} \qquad (5)$$

such that $\exists! \pi \; \exists P_O^i \in \mathbf{P_O}$ where $\pi$ is an optimal completion of $\tilde{\pi}_A$, and $\tilde{\Pi}_{P_A}$ is the set of all prefixes of a plan $\pi_A$. Plugging this scoring function in Equation 2 allows us to instantiate the design problem for predictability.

### Connection to Plan Recognition Design
The predictability problem corresponds to the plan recognition design (PRD) problem (Mirsky et al. 2019). However, our proposed framework in terms of possible observer models subsumes the plan library based approaches in being able to support a generative model of observer expectations.

## 4 Discussion and Future Work

We will now highlight limitations of the proposed framework and discuss how they may be extended in the future.

**Multiple decision making problems.** The problem of environment design, as studied in this paper, is suitable for settings where the actor performs a single repetitive task. However, our formulation can be easily extended to handle an array of tasks that the agent performs in its environment by considering a *set* of decision making problems for the actor (Sreedharan, Chakraborti, and Kambhampati 2018), where the worst-case score is decided by taking either minimum (or average) over the $wci(\cdot)$ for the set of problems.

**Interpretability Score.** The three properties of interpretable agent behavior are not mutually exclusive. A plan can be explicable, legible and predictable at the same time. In general, a plan can have any combination of the three

properties. In Equation 2, $Int(\cdot)$ uses one of these properties at a time. In order to handle more than one property at a time, one could formulate $Int(\cdot)$ as a linear combination of the three properties. In general, the design objective would be to minimize the worst-case interpretability score such that the scores for each property are maximized in the modified environment, or at least allow the designer pathways to trade off among potentially competing metrics.

**Cost of the agent.** In Section 1.3 we mentioned an advantage of the design process in the context of interpretability – the ability to offload the computational load on the actor, in having to reason about the observer model, to the offline design stage. However, there is never any free lunch. The effect of environment design is more permanent than operating on the human mental model. That is to say, interpretable behavior while targeted for a particular human in the loop or for a particular interaction, does not (usually) affect the actor going forward. However, in case of design of environment, the actor has to live with the design decisions for the rest of its life. That means, for example, if the environment has been designed to promote explicable behavior, the actor would be incurring additional cost for its behaviors (than it would have had in the original environment). This also affects not only a particular decision making problem at hand, but also everything that the actor does in the environment, and for all the agents it interacts with. As such there is a "loss of autonomy" is some sense due to environment design, the cost of which can and should be incorporated in the design process.

## Acknowledgments

This research is supported in part by the ONR grants N00014-16-1-2892, N00014-18-1-2442, N00014-18-1-2840, the AFOSR grant FA9550-18-1-0067, NASA grant NNX17AD06G and JP Morgan faculty research grant.

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
