# OpenReview forum: "Design for Interpretability"
_icaps-conference.org/ICAPS/2019/Workshop/XAIP — XAIP 2019_

### Official Review · AnonReviewer1 · 2019-05-01
**Interesting relationships between interpretability and goal/plan recognition**

**Rating:** 2
**Confidence:** 2

**Review:**

This paper is a preliminary look at the relationship between some existing definitions of interpretability and goal recognition design. The paper defines three dimensions of interpretability and relates them to goal and plan recognition design.

The ideas of interesting and well described. I think the concepts are defined (mostly) correctly and they make sense. The paper is well written and easy to read.

However, it is not clear why a focus on the relationship between interpretability and goal/plan recognition design is an important one to discuss. The definitions of making a plan legible and predictable correspond directly to the problems of goal recognition and plan recognition, so it is not suprising that there is a relationship between legible design and preictable deisng on the one hand and goal/plan recognition on the other. The relationship between goal/plan recognition and goal/plan recognition design is already clear.

I have a few recommendations for improvement:

- The small example of the coffee delivery is fine as an illustrative example, but another example that better motivates the idea of goal recongition design early on would also be useful. The authors briefly point to fixed repetitive tasks, however, for fixed, repetitive tasks, autonomy/planning is typically not required. A concrete problem that motiviates GRD would improve the paper.

- Equation 1: Why worst case intepretability instead of best case? Defining the inverse to what has been discussed throughout the paper is a bit counterintuitive.

- I think Definition 4 is incomplete? It defines a number, not a modification? Is equation 2 just meant to minimise the three expressions? If yes, this is supposed to be just minimising the summation? My intuition is that it should be saying that a DP-Int solution should minimise some weighted combination of the three expressions?

- Further, in Definition 4, it is not clear why it should minimise the *negation* of wci? This is maximising the worst case?

- In Section 3.1, why no relationship between explicability and GRD? Even if the authors believe there is none, it would be fine just note this.

---

> ### Comment · AnonReviewer4 · 2019-05-14
> **novel connection, interesting enough for a workshop?**
>
> Note: I'm not an author ;-)
>
> Personally, I was not aware before this paper that goal recognition design may connect to XAIP in the sense of design for interpretabilty/legibility/predictability. I find it a nice observation to point out this connection -- even if some of its technical aspects are rather straightforward. It's certainly worth pointing out to the community.

---

### Official Review · AnonReviewer2 · 2019-05-08
**Compelling position paper with some technical problems**

**Rating:** 2
**Confidence:** 2

**Review:**

The paper describes how an environment might be modified to increase the explicability, legibility, and predictability of plans for a given problem. The paper is a position paper that describes in general terms how these factors can be scored, and includes some discussion on issues related to modifying the environment in this way.

I'd like to see this paper accepted, but there are some technical problems, particularly in the definitions of sections 3.2 and 3.3.

"thereby minimizing the worst-case interpretability score of the actor’s cost-optimal plans for the given decision making problem."
Should this be maximizing?

The optimization function of definition 4 compares many values with possibly different weights, ie., the cost of an optimal plan, modification, explicability, legibility, etc. It would make sense that these should be at least weighted somehow before comparing the minimum score.

Definition 6 does not seem correct. The legibility is described as: "An actor’s behavior is said to be perfectly legible if it can be derived from only one model in PO. The longer it takes for a plan prefix to achieve this, the worse is the plan’s legibility." Then, lets say that pi is a prefix of plan pi_A that can be derived from at least two models (Pi,Pj) in P, where i!=j. The legibility score should be inversely proportional to the plan prefix in pi of maximum size. However, definition 6 instead talks about the minimum size, and of unique completion.

Section 3.3 is contradictory: "Therefore, predictability of a plan is inversely proportional to the length of its shortest prefix which ensures only one optimal completion solving only a single problem in the observer’s mental model." This is not how predictability was described above, ie. "the plan is predictable even though not legible". In fact, the text of 3.3 is describing both predictability and legibility together. Instead a predictable score should only be inversely proportional to the length of the shortest prefix that ensures only one optimal completion. Definition 7 must also be revised to reflect this.

The discussion and future work describes some interesting issues that will arise, particularly in the trade-off between different costs. It is a roadmap for an investigation that I would like to see.

---

> ### Author Response · Authors · 2019-05-13
> **clarifications**
>
> Thank you for the thorough review and the encouraging feedback. We have clarified a few of the questions below:
>
> Minimizing the worst-case interpretability score: Through environment design, we wish to reduce the worst case, which in turn means to improve the interpretability score. We will change the sentence to reflect that.
>
> Legibility score: For a given plan, the minimum length plan prefix, with which the plan becomes legible, is inversely proportional to the legibility score. Smaller the length of the plan prefix, higher the legibility score. For example, a plan that reveals in the first step what planning problem it is solving, has high legibility score, as against, a plan that reveals later. By unique completion, we meant that the completion in the two different models is not the same, and therefore legible.
>
> Predictability score: There is indeed only one optimal completion in predictability. Definition 7 reflects that: "exists exactly one" -- and the shortest among those is preferable.
>
> This is somewhat different from the predictability and legibility notion in existing literature (outside of XAIP in environment design). Section 2 discusses this from the XAIP perspective while later we adopt these concepts for the design problem. The last two questions from the reviewer are quite helpful: we will add a discussion to this effect on how these concepts have evolved in the context of design through a confluence of optimizing the number of possible models or plans that explain observations (in XAIP) versus the length of plan prefixes and wcd (in GRD). As it stands, the definitions stand for themselves and are inspired by existing literature, though we have not really explicated how they are different in the context of design. This would certainly improve readability and also make for a great discussion at the workshop.

---

> > ### Comment · AnonReviewer2 · 2019-05-20
> > **clarifications**
> >
> > > Predictability score: There is indeed only one optimal completion in predictability. Definition 7 reflects that: "exists exactly one" -- and the shortest among those is preferable.
> >
> > In fact I was pointing out that the definition of predictability above suggests:
> > "Therefore, predictability of a plan is inversely proportional to the length of its shortest prefix which ensures only one optimal completion solving only a single problem in the observer’s mental model."
> >
> > Should be instead:
> > "Therefore, predictability of a plan is inversely proportional to the length of its shortest prefix which ensures only one optimal completion **in any** problem in the observer’s mental model."
> >
> > That is, the plan is not yet legible, as it does not map to a single problem in the observer's mental model. However, it may still be predictable. The first definition explains this well, but Definition 7 does not.

---

> > ### Comment · AnonReviewer2 · 2019-05-20
> > **clarifications**
> >
> > I was unclear with my problem regarding Definition 6:
> > The score should be inversely proportional to the minimum length plan prefix with which the plan becomes legible. However, the equation has a problem.
> >
> > Currently it is min(plan prefix) such that there exists Pi and Pj such that the prefix can be extended to a cost optimal solution for each.
> >
> > There is two problems with this:
> >
> > 1. Consider a plan with prefixes [a0], [a0,a1], and [a0,a1,a2]. Both plan prefixes [a0] and [a0,a1] can be extended to form the optimal cost solution for more than one problem. The score should be calculated from the longer prefix [a0,a1], not [a0].
> >
> > However, definition 6 will score the problem on [a0]. Therefore definition 6 should use the **maximum** length prefix that does not correspond to a unique problem. Basically, the minimum length plan prefix with which the plan becomes legible corresponds to the maximum plan prefix with which the plan is not legible.
> >
> > 2. Definition 6 states that the prefix should have **unique** cost optimal completion. In fact, is [a0,a1] can be extended into distinct completions for different problems Pa and Pb, then the legibility score should still use [a0,a1].  (Unique completions should not be necessary, only distinct problems).

---

### Decision · Program_Chairs · 2019-05-15

**Decision:**

Accept

**Comment:**

While the reviewers view this paper critically, in the spirit of making the workshop a venue for discussion and feedback we decided to reject only those papers with strong reject votes.

Please address all review criticism as best possible for the final paper version and its presentation at the workshop. Looking forward to discuss your work at the workshop!